# Paramagnons and high-temperature superconductivity in a model family of cuprates

Lichen Wang [1,2,6], Guanhong He[1,6], Zichen Yang[2], Mirian Garcia-Fernandez [3], Abhishek Nag[3], Kejin Zhou [3], Matteo Minola [2], Matthieu Le Tacon [4], Bernhard Keimer [2], Yingying Peng [1,5✉] & Yuan Li [1,5✉]

Cuprate superconductors have the highest critical temperatures ($T_c$) at ambient pressure, yet a consensus on the superconducting mechanism remains to be established. Finding an empirical parameter that limits the highest reachable $T_c$ can provide crucial insight into this outstanding problem. Here, in the first two Ruddlesden-Popper members of the model Hg-family of cuprates, which are chemically nearly identical and have the highest $T_c$ among all cuprate families, we use inelastic photon scattering to reveal that the energy of magnetic fluctuations may play such a role. In particular, we observe the single-paramagnon spectra to be nearly identical between the two compounds, apart from an energy scale difference of ~30% which matches their difference in $T_c$. The empirical correlation between paramagnon energy and maximal $T_c$ is further found to extend to other cuprate families with relatively high $T_c$'s, hinting at a fundamental connection between them.

[1] International Centre for Quantum Materials, School of Physics, Peking University, Beijing 100871, China. [2] Max Planck Institute for Solid State Research, Stuttgart 70569, Germany. [3] Diamond Light Source, Harwell Science & Innovation Campus, Didcot, Oxfordshire OX11 0DE, United Kingdom. [4] Institute for Quantum Materials and Technologies, Karlsruhe Institute of Technology, Karlsruhe 76133, Germany. [5] Collaborative Innovation Centre of Quantum Matter, Beijing 100871, China. [6]These authors contributed equally: Lichen Wang, Guanhong He. ✉email: yingying.peng@pku.edu.cn; yuan.li@pku.edu.cn

dentifying the Cooper pairing mechanism in high-temperature superconducting cuprates is an outstanding quest in quantum materials research[1]. Magnetic interactions are widely considered to play a key role[2,3], which encourages the search for a correspondence between the Cooper pairing strength and the magnetic interaction strength. The so-called spin resonant mode[3,4] might appear to serve the purpose, but since the energy is smaller than the superconducting energy gap $2\Delta_{SC}$[4], the mode is likely a consequence of (rather than a cause for) pairing[3]. In contrast, the full spectrum of magnetic fluctuations in the form of paramagnons[5] extends to energies well above $2\Delta_{SC}$ in superconducting cuprates, making it desirable to investigate the relation between the paramagnon energy and $T_c$.

The pursuit of an experimental energy correspondence between pairing and the paramagnons has proved challenging. In the generic $p$-$T$ phase diagram[1–3] of the cuprates, where $T$ is temperature and $p$ is the doping level, the decrease of $T_c$ away from optimal doping is believed to be due to reduced superfluid density[6,7], whereas the zone-boundary paramagnon energies are nearly independent of $p$ within a given cuprate[5,8–10]. Usage of a tuning knob other than doping, such as chemical[11] and applied pressures[12], is thus required to investigate the correlation between the paramagnon energy and $T_c$. The paramagnon energy is commonly modeled and discussed in terms of the antiferromagnetic coupling strength ($J$) in the $CuO_2$ layers[5,8,10], which affects the magnetic ordering temperature ($T_N$) in parent compounds. A positive correlation between $T_c$ and $J$ inferred from $T_N$ has indeed been found previously[11]. Yet, spectroscopic determination of the associated energies suggested a weaker correspondence[13], and historically different impressions were once obtained[12,14–16]. This is because major modifications of $J$ are difficult to achieve, and they go along with structural modifications whose consequences are difficult to assess. Moreover, in materials with relatively low $T_c$, it is unclear whether the variation in $T_c$ should be considered correlated with $J$, or caused by detrimental effects on $T_c$ including chemical disorder[17–19] and competing states[1], which may vary at the same time as the tuning takes place. These limitations have prevented the previous experimental indications from being widely recognized as having established a conclusive connection between the pairing and the magnetic energies.

To advance on this front, a desirable approach is to study materials with very high yet sufficiently different $T_c$, so that $T_c$ is not strongly reduced by material-specific details and its variation can be more reliable compared to that of the magnetic energy. We further aim to study chemically similar materials with similar doping and to determine $T_c$, $2\Delta_{SC}$, and paramagnon energy using the same samples. The materials of our choice are the first two Ruddlesden-Popper (RP) members of the Hg-family of cuprates, $HgBa_2CuO_{4+\delta}$ (Hg1201) and $HgBa_2CaCu_2O_{6+\delta}$ (Hg1212). Hg1201 and Hg1212 are chemically nearly identical, have the highest $T_c$ among single- and double-layer cuprate families (97 and 127 K at optimal doping, respectively)[17] and all $CuO_2$ layers are identical by symmetry. For later RP members with three or more consecutive $CuO_2$ layers, the charge imbalance between the inequivalent layers[20] complicates the analysis. Previously, the difference in $T_c$ between Hg1201 and Hg1212 has been attributed to quantum tunneling of Cooper pairs between the adjacent $CuO_2$ layers[20]. But as we will show, a variation in the magnetic energy within the individual $CuO_2$ layers appears empirically sufficient to account for the differences in $T_c$ and $2\Delta_{SC}$.

## Results

**Resonant inelastic x-ray scattering result**. Benefiting from the recent breakthroughs in crystal growth[21,22], our experiments are performed on nearly equally underdoped high-quality single crystals of Hg1201 and Hg1212 (Supplementary Fig. 1), with $T_c$ of 80 K ($p \sim 0.11$) and 107 K ($p \sim 0.12$), respectively. Our primary measurement technique is resonant inelastic x-ray scattering (RIXS). By using incident x-ray photons tuned to the energy of the $L_3$ absorption edge of $Cu^{2+}$, RIXS directly probes paramagnons in doped cuprates with ever-improving resolution and signal quality[5,23]. The dominance of these collective excitations over incoherent particle-hole excitations in the RIXS spectra has been demonstrated up to at least optimal doping $p \sim 0.16$ in various cuprates[24,25]. Besides, we have also used RIXS to confirm that the crystal field levels of $Cu^{2+}$ are nearly identical between Hg1201 and Hg1212 (Supplementary Fig. 2).

Figure 1 displays our representative RIXS spectra obtained with $\pi$-polarized incident photons along the high-symmetry lines $\mathbf{Q}_{//} = (H, 0)$ and $(H, H)$ of the magnetic Brillouin zone. The intensity consists of five components: elastic, single- and two-phonon scattering, a weakly energy-dependent background, and magnetic scattering mainly from paramagnons[5,8]. Our model fitting satisfactorily accounts for the measured intensity (Fig. 1a, b, e, f, see Methods for details), where the paramagnon signal is described by a damped harmonic oscillator (DHO) peak visualized by shaded areas in Fig. 1. This peak clearly disperses along the $(H, 0)$ direction (Fig. 1c, g), and a comparison between the two systems close to the Brillouin zone boundary at $H = 0.46$ indicates a distinct energy increase from Hg1201 (Fig. 1a) to Hg1212 (Fig. 1e), which amounts to $22 \pm 9\%$ of the energy in Hg1201 as determined from the fit maximum of the DHO peak. Albeit overdamped, paramagnons in the $(H, H)$ direction (Fig. 1b, f) are consistent with a similar energy increase from Hg1201 to Hg1212.

The above observation calls for a systematic comparison of the paramagnon spectra of the two systems, which we present in detail in Supplementary Figs. 3–6 and Supplementary Tables 1, 2. Our conclusion is that, from Hg1201 and Hg1212, the propagation energies of the paramagnons[5] increase globally by approximately 30%. To visualize this, we plot in Fig. 2 the energy- and momentum-dependent RIXS intensities, after removing the non-magnetic contributions and normalizing the integrated spectral weight at each $\mathbf{Q}_{//}$. For Hg1212 (Fig. 2b, d), the displayed vertical energy range is purposely set to be 130% of that for Hg1201 (Fig. 2a, c). A comparison in the same energy range can be found in Supplementary Fig. 7. It is clear that the results are very similar apart from the energy-scale difference. A pursuit for the best visual similarity would instead suggest a rescaling ratio of about 122% (Supplementary Fig. 8), close to the comparison in Fig. 1a, e. Since visual comparison may be biased by the effect of damping, here we consider the ratio (~130%) associated with the propagation energy near the zone boundary (0.5, 0) physically more meaningful, because the propagation energy there is experimentally and theoretically known to change little with doping[5,8–10,26], so it plausibly reflects the intrinsic magnetic energy scale for a given cuprate.

Importantly, the 22–30% increase in the paramagnon energy from Hg1201 to Hg1212 is enough to correlate with most, if not all, of the difference in $T_c$ between the two systems. Although magnon branch-splitting is not resolved here in Hg1212 (see Methods), the magnetic coupling between the adjacent $CuO_2$ layers in a bi-layer cuprate is known to be only about 10 meV, which has little effect on the (para)magnon energy far away from the zone center[27]. Thus, the observed energy increase must originate in the strength of interactions within the $CuO_2$ layers. Indeed, if one takes $T_c$ as the measure for the pairing strength, our result readily suggests a near-proportional correlation between the pairing and the magnetic energies.

**Raman scattering result**. The pairing energy may also be gauged by measuring the superconducting gap $2\Delta_{SC}$. To determine $2\Delta_{SC}$,

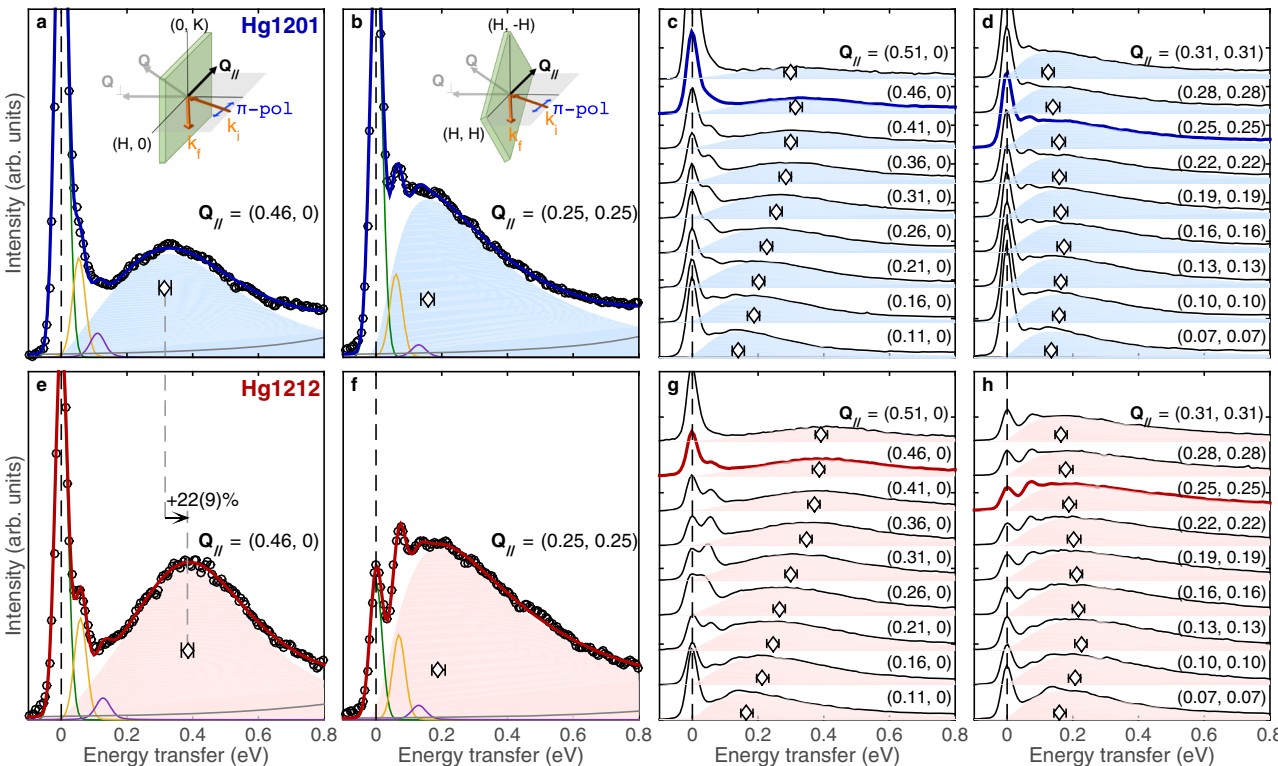

**Fig. 1 RIXS spectra measured at $T = 13$ K with $\pi$-polarized incident x-rays. a, b** Representative data measured on Hg1201 with in-plane momentum transfers $\mathbf{Q}_{//} = (0.46, 0)$ and $(0.25, 0.25)$, respectively, in reciprocal lattice units (r.l.u.). The data are normalized to the intensity of *dd* excitations at higher energies (Supplementary Fig. 2), and then fitted to a sum of an elastic peak (green), a single-phonon peak (yellow), a two-phonon peak (magenta), a paramagnon peak (shaded area, the diamond symbol indicates the peak maximum), and a weakly energy-dependent background (gray). Details of the fitting are presented in Methods, and the results are shown in Supplementary Figs. 3–6 and Supplementary Table 1. Insets illustrate the scattering geometry, where the detector is placed at a fixed 2θ angle of 154° from the incident beam. The desired $\mathbf{Q}_{//}$ is reached by rotating the sample around the vertical axis. **c, d** Spectra at a series of $\mathbf{Q}_{//}$ along high-symmetry directions, vertically offset for clarity. The fitted paramagnon peaks are displayed by shaded areas along with the data. **e–h** Same as **a–d**, but for Hg1212. Error bars represent uncertainty in the estimate of the energies of the paramagnon signal's intensity maxima (1 s.d.), see Methods for details.

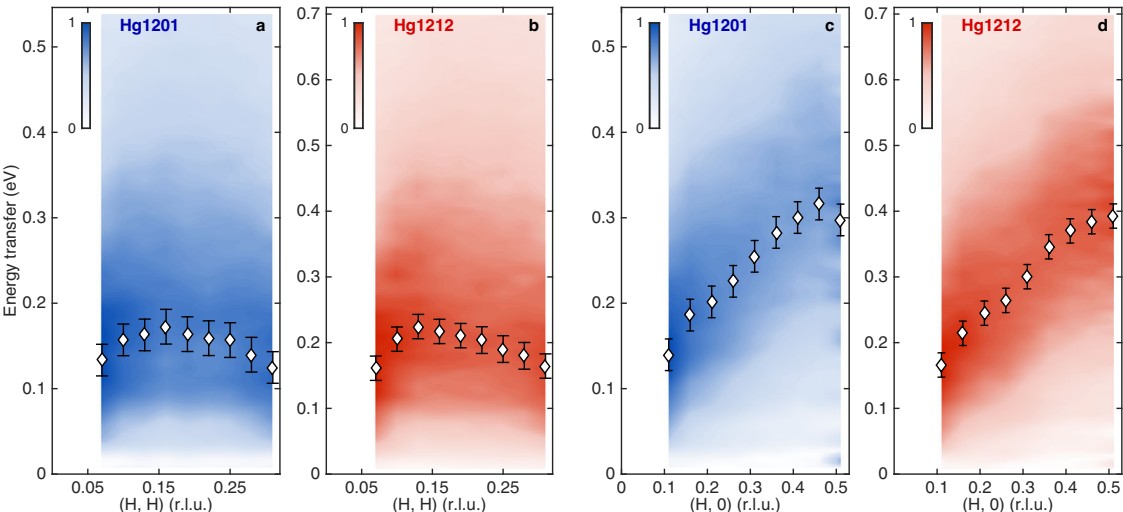

**Fig. 2 Comparison of paramagnons in Hg1201 and Hg1212. a, b** False-color representation of energy- and momentum-dependent RIXS intensities arising from paramagnons in Hg1201 and Hg1212, respectively, along the $(H, H)$ momentum direction. The signal is extracted from the data displayed in Fig. 1d, h, after subtracting the non-magnetic contributions and normalizing the energy-integrated spectral weight among different $\mathbf{Q}_{//}$. Diamond symbols indicate energy positions of intensity maxima (according to the DHO fits in Fig. 1) at the measured $\mathbf{Q}_{//}$, with values and error bars identical to those in Fig. 1. **c, d** Same as **a, b**, but for the $(H, 0)$ momentum direction. For visual comparison between the two systems, the vertical energy scales of **b** and **d** are set to be 130% of those of **a** and **c**.

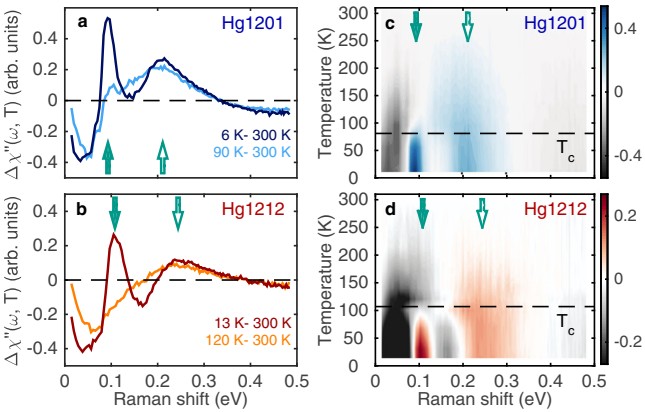

**Fig. 3 $B_{1g}$ Raman spectra for Hg1201 and Hg1212. a**, **b** Bose-factor corrected electronic Raman spectra relative to 300 K for Hg1201 and Hg1212, respectively. The $B_{1g}$ scattering geometry involves incident and scattered photons linearly polarized along the diagonals of the CuO$_2$ plaquettes, and perpendicular to each other. **c**, **d** False-color representation of Raman spectra at various $T$ relative to 300 K. Solid arrows indicate the energy of the pair-breaking peak, at 93 (108) meV for Hg1201 (Hg1212), and empty arrows the two-paramagnon peak, at 211 (244) meV for Hg1201 (Hg1212), estimated from spectra obtained at the lowest temperature. Additional data are presented in Supplementary Fig. 9.

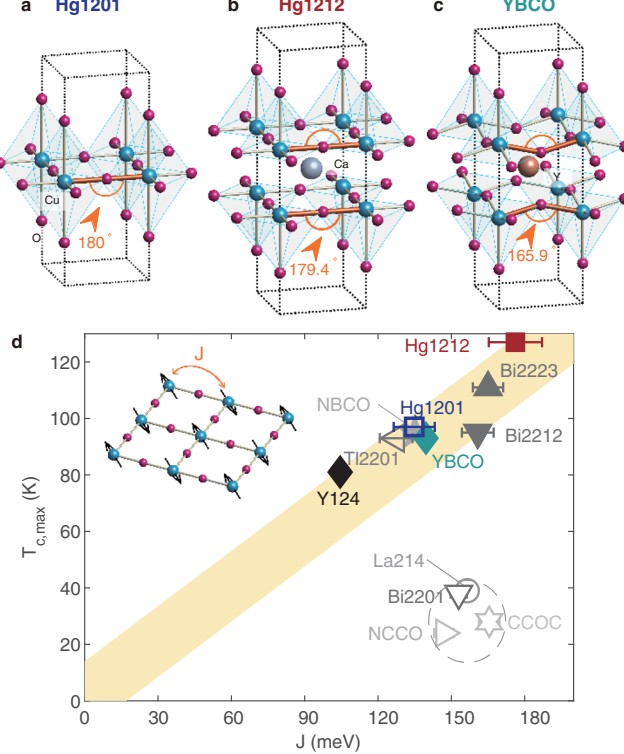

**Fig. 4 Bonding geometry of CuO$_2$ layers and summary of $T_{c,max}$ versus $J$ in different cuprates. a–c** Schematics of CuO$_2$ layers (with apical oxygens) of Hg1201, Hg1212, and YBCO. The Cu-O-Cu bond angle (see text) is indicated in orange, and values of other compounds are summarized in Supplementary Table 3. **d** $T_c$ of various cuprates at optimal or the best available doping, plotted versus $J$ extracted from neutron scattering and RIXS, see Supplementary Table 4 for values and references. The thick yellow line is a guide to the eye, and its upper boundary implies an upper bound for $T_{c,max}$ discussed in the text. Strong outliers from this line are enclosed by a dashed ellipse. Inset illustrates $J$ between neighboring Cu$^{2+}$ across ligand oxygen atoms.

we have performed variable-temperature $B_{1g}$ electronic Raman scattering[28] on the same two crystals studied by RIXS. Figure 3 displays our Raman spectra taken at low temperatures after subtracting their room-temperature references, which highlights the appearance of the superconducting pair-breaking peak (at the energy of $2\Delta_{SC}$) below $T_c$. It is found that the increase in $2\Delta_{SC}$ from Hg1201 to Hg1212 in our samples is about 16%, considerably smaller than the 34% increase in $T_c$. This difference may be partly attributed to an enhancement of $T_c$ (but not necessarily $2\Delta_{SC}$) in bi-layer Hg1212 by the inter-layer coupling[20], and to a possible slight difference in doping of our crystals since $T_c$ and $2\Delta_{SC}$ are known to vary disproportionally with underdoping[29,30]. Near-optimal doping, $2\Delta_{SC}$ is known to be about 86 meV (ref. [30], compared to 93 meV in our sample) for Hg1201; while no measurement of $2\Delta_{SC}$ has been reported for optimally doped Hg1212, we expect it to be somewhat smaller than 108 meV in our underdoped sample (Fig. 3b). Therefore, we estimate the increase in $2\Delta_{SC}$ (from Hg1201 to Hg1212, at optimal doping or the same doping) to be between 16 and 26%. The 22–30% increase in the paramagnon energy is again very comparable to it.

In addition to the pair-breaking peak, a broad Raman peak is found to develop at higher energy than $2\Delta_{SC}$ upon cooling (Fig. 3, see Supplementary Fig. 9 for additional data). This peak arises from excitations involving the interchange of spins (i.e., double spin-flip with total $\Delta S = 0$[10]) on Cu$^{2+}$, and is known as the bi-(para)magnon peak[28]. While the energy of the peak depends on doping due to additional effects[10,30,31] and cannot be used to unambiguously determine the strength of magnetic interactions, it has been reported that the bi-paramagnon energy approximately tracks the doping evolution of $2\Delta_{SC}$ over a substantial range[29–32] and, upon cooling, the peak intensity increases concurrently with the formation of Cooper pairs[29]. Our data in Fig. 3 reaffirm and extend these findings: the bi-paramagnon peak grows and becomes better-defined upon cooling into the superconducting state in both Hg1201 and Hg1212; between the two systems, despite the fact that the relative difference (16%) in the peak energy determined from the spectral variation with temperature is smaller than the RIXS result, it matches precisely that of $2\Delta_{SC}$. We believe that the intriguing empirical

correspondence warrants further study, especially in the light of our RIXS results.

## Discussion

**The effect of crystal and electronic structures on $J$.** An analysis of the crystal and electronic structures is helpful at this point. As a bi-layer cuprate, Hg1212 has considerably higher paramagnon energy than YBa$_2$Cu$_3$O$_{6+\delta}$ (YBCO)[5], which has been correctly predicted by ab initio calculations[33] and is in accordance with the straighter Cu-O-Cu bonds in Hg1212 than in YBCO (Fig. 4b, c). The higher paramagnon energy than in Hg1201, however, is unexpected in the calculations where the on-site Coulomb repulsion is set constant[33], and it seems to arise from a variation in the electronic wave functions (and hence the on-site repulsion)[34]. Experimentally, the charge-transfer gap has been observed to decrease significantly from single- to bi-layer cuprates[35], probably due to differences in the apical ions (Fig. 4a, b). The charge-transfer gap may play a similar role as the on-site repulsion and be inversely proportional to $J$[2,35,36]. Our result implies that Hg1212 has a smaller charge-transfer gap than Hg1201 in their parent compounds. The reversed question may also be asked: With a smaller charge-transfer gap[35], why does Bi$_2$Sr$_2$CaCu$_2$O$_{8+\delta}$ (Bi2212) not have substantially larger paramagnon energy than Bi$_2$Sr$_{2-x}$La$_x$CuO$_{6+\delta}$ (Bi2201)[15,16]? The answer, as we detail in Supplementary Table 3, plausibly lies again

in the crystal structure—Bi2212 suffers from a greater departure of the Cu-O-Cu bond angle from 180° than Bi2201. Besides, we note that a recent calculation based on the three-band Hubbard model indicates that electron covalency can increase $J$ for a given charge-transfer gap[36]. It is, therefore, reasonable to believe that $J$ can vary considerably according to the crystal and electronic structures. Since $J$ is the leading magnetic interaction term, this provides a rationale for our approximate description of the paramagnon spectra using a single parameter $J$.

**Comparison among additional cuprates**. We now turn to a broader discussion comparing our result to additional cuprates. Using $J$ as a simplified and unified description of the paramagnon energy measured with inelastic neutron scattering and RIXS (detailed in Supplementary Fig. 10), we summarize the result and the different families' maximal value of $T_c$ ($T_{c,max}$, at optimal or the best available doping) in Fig. 4d. The quoted values of $J$, obtained by linear spin-wave theory fitting (see Methods), are mainly constrained by the observed energies of zone-boundary (para)magnons, and a similar trend between the latter and $T_{c,max}$ is shown in Supplementary Fig. 11. This direct spectroscopic origin of the extracted $J$ values renders it a simplified yet reliable quantity for comparison among the different families, some of which (including Hg1201 and Hg1212) lack parent compounds. Importantly, our new Hg1201 and Hg1212 result extend the data range both horizontally and vertically in Fig. 4d, revealing an approximately linear relationship between $T_{c,max}$ and $J$ that includes a total of eight different cuprates with relatively high $T_{c,max}$. The empirical observation is consistent with a former study using chemical substitution to tune both $J$ (estimated from $T_N$ and inter-layer coupling) and $T_c$[11], but a broader data range and many more compound families are covered here.

The cuprates are well-known to have the Cooper pairs formed without phase coherence above $T_c$ due to the low superfluid density[6,7]. The pair formation temperature ($T_{pair}$) above $T_c$ is hence an alternative measure of the pairing strength. However, values of $T_{pair}$ vary widely in the literature depending on the method of observation: by tracing the temperature evolution of gap formation (presumably due to pairing), angle-resolved photoemission spectroscopy revealed $T_{pair}$ of Bi2201 and Bi2212 to be between 120 to 150 K in spite of their very different $T_c$[37]. Other techniques also supported similar $T_{pair}$ in the Bi-family, YBCO and La214[38–40]. According to these results, the above compounds have similar $T_{pair}$ (even though some of them have distinct $T_c$), and we note that they also possess similar $J$ in Fig. 4d. In contrast, torque magnetometry suggested a universal and rather narrow temperature range of superconducting fluctuations (only a few Kelvins) above $T_c$[41,42]. In this definition, $T_{pair}$ would closely track $T_c$, so the maximal $T_{pair}$ of a given family would have a similar relation to $J$ as $T_{c,max}$. We, therefore, conclude that despite on-going debates about $T_{pair}$ in the literature (see Supplementary Table 5), the relationship between $T_{pair}$ and $J$ can be expected to remain consistent with the empirical trend revealed in Fig. 4d.

Admittedly, there are distinct outliers from the linear trend in Fig. 4d, hence one should use cautions and not prematurely interpret the trend as suggesting a causality relation between $J$ and $T_{c,max}$. In fact, without our Hg-family data points, the approximate linear trend would have become much less evident even among the high-$T_{c,max}$ cuprates. Specifically, considering the data points of single- and bi-layer cuprates including $Tl_2Ba_2CuO_{6+\delta}$ (Tl2201), YBCO, $YBa_2Cu_4O_8$ (Y124), $NdBa_2Cu_3O_{6+\delta}$ (NBCO), and Bi2212, one might obtain the impression that $J$ can vary by over 50% without affecting $T_{c,max}$ by more than 10%. Some of these compounds' further comparison to the low-$T_{c,max}$

ones including Bi2201, $La_{2-x}(Sr,Ba)_xCuO_4$ (La214), $Ca_{2-x}Na_xCuO_2Cl_2$ (CCOC), and $Nd_{2-x}Ce_xCuO_4$ (NCCO) could even suggest an anti-correlation between $J$ and $T_{c,max}$. Meanwhile, a comparison within the bismuth-family of cuprates, Bi2201, Bi2212, and $Bi_2Sr_2Ca_2Cu_3O_{10+\delta}$ (Bi2223), could suggest that $T_{c,max}$ increases by a factor of three without much variation in $J$, which would appear to indicate that the number of multiple $CuO_2$ layers in the primitive cell is much more relevant to the variation in $T_{c,max}$ than the strength of magnetic interactions. Our Hg1201 and Hg1212 data points are thus highly elucidating in such an elusive situation, as the Hg-family is believed to suffer the least from material-specific detrimental factors on $T_c$[17–19], which makes them more likely to reveal intrinsic correlations.

We end our discussion with a speculative note motivated by the outliers in Fig. 4d. Their relatively low $T_{c,max}$ could be attributed to detrimental effects on $T_c$ including disorder[17], competing order[1] (e.g., stripe order in La214), and a small hopping range of conduction electrons[43,44]. Meanwhile, the reason behind the outliers might be related to a recent proposal that there is an optimal coupling strength between charge carriers and pair-mediating bosons in weak-coupling superconductors[45,46], where too strong couplings promote charge-order formation rather than superconductivity. We notice that La214, Bi2201, and NCCO have been reported to exhibit pronounced anomalies in their paramagnon spectra near a wave vector where charge correlations have been observed[16,47,48]. Such anomalies hint at strong couplings between charge carriers and magnetic excitations in those systems[49,50], yet similar anomalies are nearly invisible in the Hg1212 and Hg1201 (Fig. 2d and ref. [51]). Whether this hints at a moderate coupling strength in Hg1201 and Hg1212 which favors high $T_c$ warrants further experimental and theoretical studies.

In summary, we report on an empirical correlation between paramagnon energy and $T_c$ in superconducting samples of Hg1201 and Hg1212. Our finding brings fresh insight into the discussion of the role of antiferromagnetic coupling strength in affecting $T_c$ in the cuprates[36,52], and renders the Hg-family of cuprate an ideal platform to investigate the intrinsic pairing mechanisms by state-of-the-art spectroscopic methods.

## Methods

**Sample preparation and characterization**. The Hg1201 and Hg1212 single crystals used in this study were grown with a self-flux method[21,22]. Photos of the crystals are displayed in Supplementary Fig. 1a. The crystals were post-growth annealed over extended periods of time in air at 480 °C, in order to reach homogeneous doping as indicated by their sharp transitions at $T_c$ (Supplementary Fig. 1b) determined from magnetometry (Quantum Design MPMS VSM). The resultant doping levels are estimated[53] to be $p \sim 0.11$ for Hg1201 and 0.12 for Hg1212 based on their $T_c$ values of 80 and 107 K, respectively, using a simplified relation:

$$T_c = T_{c,max}(1 - 82.6 \times (p - 0.16)^2) \qquad (1)$$

The good crystallinity is demonstrated by single-crystal x-ray diffraction (Rigaku MiniFlex 600) and x-ray Laue diffraction (Photonic Science, Supplementary Fig. 1a, c). For both the RIXS and Raman measurements, the crystals were freshly polished along their $ab$ plane with 0.05 μm-grade 3 M lapping films before being loaded into the vacuum.

**RIXS experiment**. The RIXS experiments were performed at beamline I21 of Diamond Light Source, Didcot, United Kingdom. The incident x-ray energy was tuned to the $L_3$ absorption edge of $Cu^{2+}$ at about 931.5 eV and was calibrated frequently during the experiments by performing x-ray absorption spectroscopy measurements in the total fluorescence yield mode. The beam size on a sample with full flux was $40(H) \times 2.5(V)$ μm². The total instrumental bandwidth (energy resolution) at the Cu $L_3$ absorption edge was about 37 meV, determined as the full width at half-maximum (FWHM) of the diffuse scattering peak from a carbon tape mounted at the sample position. All RIXS spectra were collected at a temperature of about 13 K. With the exception of some data in Supplementary Fig. 13, all RIXS spectra were obtained using π-polarized incident x-rays for maximal sensitivity to single spin-flip excitations. The detector was placed at 154°, and the polarization state of the scattered phonons was not analysed. For this reason, overwhelming charge-scattering dominations (the elastic line) were observed in the spectra close

to (0, 0) where too large uncertainties were obtained for the fitting results of paramagnon, so we only display scans at $Q_{//}$ larger than (0.11, 0) and (0.07, 0.07). For the bi-layer compound, the optical and acoustic magnon overlap away from the zone center[5], therefore the magnon splitting of two branches in Hg1212 cannot be resolved in this setup. The raw data are displayed in Supplementary Fig. 2, with momentum coverage along two high-symmetry directions of the first magnetic Brillouin zone. The lattice parameters we used for calculating the scattering geometry were $a = b = 3.840$ Å, $c = 9.435$ Å for Hg1201, and $a = b = 3.788$ Å, $c = 12.557$ Å for Hg1212.

**Raman scattering experiment**. The Raman scattering experiments were performed in a confocal back-scattering geometry using a Horiba Jobin Yvon Lab-RAM HR Evolution spectrometer equipped with 600 lines/mm grating, a liquid-nitrogen-cooled CCD detector, and a He-Ne laser with $\lambda = 632.8$ nm as the excitation line. During the measurements, the samples were kept in a liquid-helium flow cryostat (ARS) under an ultrahigh vacuum (~$10^{-8}$ torr), and all data were obtained in the $B_{1g}$ scattering geometry[28], with the incident and the scattered photons linearly polarized perpendicular to each other and along the diagonals of the $CuO_2$ plaquettes. The laser power on the sample was kept below 0.65 mW, thereby avoiding heating effects.

The Bose-factor corrected Raman spectra are displayed in Supplementary Fig. 9a, b. The data have been corrected for the optical response of the measurement system and normalized around 0.33 and 0.38 eV Raman shift for Hg1201 and Hg1212, respectively. Defect phonon peaks ranging from 450 to 650 cm$^{-1}$ [22] have been removed from the spectra to focus the attention on the electronic Raman scattering signal. Both the pair-breaking peak and the bi-paramagnon peak become most evident in the data taken at low temperatures after subtracting the 300 K spectrum as reference (Supplementary Fig. 9c, d). However, the bi-paramagnon peak is already present as a broad hump at 300 K. Thus, in this way, the bi-paramagnon peak that we present in Fig. 3 should be regarded as the temperature-dependent part of the bi-paramagnon signals.

**Analysis of RIXS spectra**. To facilitate a systematic analysis and comparison of the RIXS spectra, we first normalize the spectra taken at different $Q_{//}$ to the intensity of the $dd$ excitations[54] (from 1 to 3.5 eV, based on data in Supplementary Fig. 2). The normalized data acquired at different $Q_{//}$ for Hg1201 and Hg1212 are then compared in Supplementary Figs. 3 and 4.

We describe the RIXS intensities below 1 eV with a total of five spectral components: a resolution-limited elastic peak, a resolution-limited single-phonon peak, a weakly resolution-limited two-phonon peak, a paramagnon peak, and a weakly energy-dependent background. The resolution-limited components are modeled by Gaussian peaks of fixed FWHM of 37 meV, and the weakly resolution-limited component is described by convolving the Gaussian peak with a Lorentzian peak of smaller FWHM than the Gaussian peak. The background component is modeled by a Lorentzian peak centered at the energy of the $dd$ excitations (the background is just the tail of this peak). The paramagnon component is described by a generic damped harmonic oscillator $L(\omega)$ convolved with the Gaussian resolution function,

$$L(\omega) = \frac{\gamma\omega}{\left(\omega^2 - \omega_0^2\right)^2 + 4\gamma^2\omega^2} \tag{2}$$

where $\omega_0$ is the undamped frequency and $\gamma$ is damping. When $\gamma < \omega_0$, this function can be identically reproduced as an anti-symmetrized Lorentzian peak for $\omega > 0$,

$$L(\omega) = \frac{1}{4\omega_p}\left(\frac{\gamma}{(\omega - \omega_p)^2 + \gamma^2} - \frac{\gamma}{(\omega + \omega_p)^2 + \gamma^2}\right) \tag{3}$$

where the propagation frequency $\omega_p^2 = \omega_0^2 - \gamma^2$. Therefore, $\omega_p$ lacks definition when $\gamma > \omega_0$, as is the case for $Q_{//}$ along $(H, H)$ due to the overdamped nature of the paramagnons (Supplementary Fig. 5 and Supplementary Table 1).

The fitting procedure first requires fixing the individual spectrum's zero energy according to the center position of the elastic peak, which in subsequent iterations of the fitting is set to zero. The model parameters are then determined by the least-square method, where most parameters are considered momentum-dependent and free to vary, except for certain constraints on the inessential parameters concerning the two-phonon peak and the background. Specifically, the two-phonon peak energy is assumed to be independent of $Q_{//}$ because it is found to be weakly dispersive. The peak position of the Lorentzian-tail-like background was fixed to the energy of the $dd$ excitation[55]. The resultant best-fit parameters concerning the paramagnon component, and the associated comparison between Hg1201 and Hg1212, are presented in Supplementary Figs. 5, 6 and Supplementary Tables 1, 2. We note that while our RIXS data for Hg1201 are consistent with those in a recent report on Hg1201[51] wherever a direct comparison can be made (Supplementary Fig. 12), details of the analyses might be different. By using the same method to analyse both Hg1201 and Hg1212, we are able to minimize systematic errors concerning the quantitative comparison between the two compounds.

Because our RIXS data are of very high statistical accuracy and energy-sampling density, the accuracy of model-parameter estimation, especially on the parameters concerning the paramagnon signal, is not limited by the data quality but rather by the accuracy of the model. Therefore, given that the first step of our fitting involves

a self-correction of the zero-energy reference point using the resolution-limited elastic peak, whenever the fitting uncertainty on the paramagnon energy parameters ($\omega_0$ or $\omega_p$, which in turn determines the maximal-intensity energy $\omega_{max}$) is smaller than 19 meV, the half-width at half-maximum of the resolution function, we consider the uncertainty to be 19 meV. In addition, we have found that the background amplitude can affect the estimation of $\omega_{max}$, $\omega_0$, $\gamma$, and $\omega_p$, hence we estimate the size of their confidence range by manually fixing the background amplitude to its allowable maximum according to the data and observing how the fit results vary. In this way, we conclude that we have considerably larger uncertainty in the determination of $\omega_0$ and $\gamma$ along $(H, H)$ than $(H, 0)$ (Supplementary Fig. 5), and also for $\omega_p$ at smaller $H$ along $(H, 0)$ (Supplementary Fig. 6). This result is generally consistent with previous RIXS results for doped cuprates, e.g., in ref. [8]. The fitted value of $\omega_p$ exhibits a kink-like structure in its dispersion between $H = 0.26$ and 0.3 along $(H, 0)$, which is resulting from the change of $\omega_0/\gamma$ ratio in this momentum range (Supplementary Fig. 5). As this momentum range corresponds to short-range charge correlations in the Hg-family of cuprates[51,56–58], as is also suggested by the elastic-peak intensity in our RIXS data (Supplementary Fig. 13), the result is consistent with the notion that there is an interplay between the charge and magnetic correlations and the associated properties of their excitations[47]. We emphasize that, while we are able to notice this anomaly via fitting, the anomaly is nearly invisible in the intensity spectrum (Fig. 2c, d) and is therefore a weak effect in the Hg-family of cuprates.

**Extraction of $J$ for different cuprates**. In order to compare $J$ among different cuprates, we consider a nearest-neighbor-coupled spin-1/2 Heisenberg model on a square lattice:

$$H = J \sum_{\langle i,j \rangle \in NN} S_i \cdot S_j \tag{4}$$

where $J$ is the Heisenberg antiferromagnetic interaction between the nearest neighbors. In our case of doped Hg1201 and Hg1212, the paramagnon propagation energy $\omega_p$ lacks its definition along $(H, H)$ due to the overdamped nature of the RIXS signals. Thus, we only consider $\omega_p$ dispersion along $(H, 0)$ for the extraction of $J$ for other cuprates as well, in order to maintain the most consistent standard. In our model, the neglection of high-order interactions (for instance, cyclic exchange $J_c$) might lead to a slight underestimation of the real nearest-neighbor exchange in CCOC and La214[44], but a correction to those data points in Fig. 4 will not affect our conclusion. With the understanding that $J_c$ is reflected by the energy difference between (0.5, 0) and (0.25, 0.25)[44], it is expected to be similar between our superconducting samples of Hg1201 and Hg1212 (Supplementary Table 1), yet we note that the most accurate value of $J_c$ should be obtained from measurements of undoped samples which are not available at present. Using the linear spin-wave theory (LSWT), the dispersion of the paramagnon energy can be simply written as

$$\omega(H) = 2J\sqrt{1 - \frac{(\cos(2\pi H) + 1)^2}{4}} \tag{5}$$

where $H$ is the value in $Q_{//} = (H, 0)$ in units of r.l.u.

As doping increases, the (para)magnon signal becomes broadened in energy, but the high-energy part of the spin excitations near the zone boundary (0.5, 0) have been demonstrated to hardly change, both experimentally[8,9,26,59] and theoretically[10], compared to the parent compound. We, therefore, consider it physically reasonable to rely on reported values of $\omega_p$, available for La214[60], CCOC[61], YBCO[9], Y124[5], NBCO[5,44], Tl2201[9], Bi2201[8], Bi2212[15,16], Bi2223[15], NCCO[62] for the extraction of $J$. For parent compounds, we use $\omega_p$ at all measured momenta along $(H, 0)$. For doped systems, since spin excitations hold collective nature up to the optimal doping[25] and its high-energy dispersion in the underdoped sample is similar to that of the parent compound even with the damping[8,47], the LSWT is considered still valid to fit the high-energy paramagnon in underdoped samples. Therefore, we only use $\omega_p$ at $Q_{//} \geq (0.3, 0)$ where the influence of damping, manifested as the departure of $\omega_p$ from $\omega_{max}$, is relatively small. Our fitting of the published data is presented in Supplementary Fig. 10, and the extracted values of $J$ are summarized in Supplementary Table 4 with uncertainty estimated based on the fits. Alternatively, we have also attempted to compare $T_c$ directly to the (para)magnon energies near (0.5, 0) (denoted as $\omega_{p,max}$), which are summarized in Supplementary Table 4 and plotted in Supplementary Fig. 11.

**Structural properties, $T_{c,max}$ and $T_{pair}$ of different cuprates**. In Supplementary Table 3, we summarize some key aspects of structural properties, along with their values of $T_{c,max}$ and most prominent disorder site (when doped). The associated structural data were originally reported in refs. [17,63–82]. Materials with higher $T_{c,max}$ are generally observed to be those with larger Cu-O-Cu angle[63], larger Cu-O apical distance[63], weaker structure disorder[17], and larger hopping ranges[43,44]. In Supplementary Table 5, we summarize $T_{pair}$ measured by various techniques. Their values are extracted from refs. [37–40,42,83–91].

## Data availability
All data are available in the main text or the Supplementary Information.

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

## Acknowledgements

We are grateful to D. Betto, A. Chubukov, Ji Feng, M. Greven, A. Henri, E. Huang, G. Khaliullin, H.-H. Kim, S. A. Kivelson, H. Suzuki, Y. Lu, T. Valla, Xiangang Wan, Fa Wang, Yayu Wang, and Yuanbo Zhang for discussions. We thank Jiarui Li and Xiangpeng Luo for their help at the early stage of the project and Prof. Shuang Jia for access to their XRD apparatus. Y.L. is grateful for financial support from the National Natural Science Foundation of China (NSFC, Grants No. 12061131004 and No. 11888101) and the National Key Research and Development Program of China (Grants No. 2018YFA0305602 and No. 2021YFA1401900), and for the Tsung-Dao Lee Institute at Shanghai Jiao Tong University for hospitality. Y.P. is grateful for financial support from the National Natural Science Foundation of China (Grant No. 11974029) and the National Key Research and Development Program of China (Grant No. 2019YFA0308401). L.W. is partly supported by the Alexander von Humboldt Foundation.

## Author contributions

L.W. and Y.L. conceived the research. L.W., M.M., M.L.T., Y.P., and Y.L. designed the experiments. G.H. and L.W. prepared the samples. L.W., G.H., Z.Y., and Y.P. performed the RIXS experiment with the help of M.G.-F., A.N., and K.Z. at the Diamond Light Source. L.W. performed preliminary Raman experiments and G.H. performed Raman measurements on the crystals studied with RIXS. L.W. and G.H. analysed the experimental data and analysed published results on magnetic excitations and crystal structure of other cuprates. M.M., M.L.T., B.K., Y.P., and Y.L. advised and oversaw the project. L.W., G.H., and Y.L. wrote the manuscript with input from all co-authors.

## Competing interests
The authors declare no competing interests.
