## [Peer Review File · Nature Communications]

Reviewers' Comments:

Reviewer #1:

Remarks to the Author:

I have read the manuscript "Paramagnons and high-temperature superconductivity in a model family of cuprates" by Wang et al. with interest. The authors report an X-ray and Raman scattering study of the energy scales of the spin excitations and superconducting gap in the family of Hg cuprates. The authors purport that the observed correlation between the paramagnon energy, and correspondingly the Cu-Cu nearest-neighbor exchange interaction J , and the superconducting critical temperature is indicative, or perhaps strongly suggestive, of spin-mediated Cooper pairing in the cuprates.

Needless to say, what is the nature of the excitations and interactions mediating the formation of Cooper pairs in copper oxide high- T_c 's is a long-standing question that is central to the understanding of unconventional superconductivity. The authors sought to provide fresh insights to this question. They adopt an approach which has been followed in prior inelastic neutron scattering studies of spin excitations in the cuprates – that is, modeling the spin dynamics to extract an effective magnetic interaction scale (J) to compare to energy (Δ_{SC}) or temperature (T_c) scales relevant for superconductivity. The main novelty is that they focus on two cuprate families (Hg1201 and Hg1212) that had not been investigated very systematically and finish to compile the new data together with historical data to extract trends and correlations, especially between J and $T_{c,max}$.

I enjoyed reading the study and learning about the new analysis of RIXS data in the Hg-cuprate family.

I personally think that the study would be worth of publication in Nature Communications on the sole standing of the present data, without invoking broad conclusions which, to my reading, sound somewhat speculative. I would be happy to endorse (after revisions – see remarks below) this manuscript for publication if the authors take this route. However, I cannot support publication of this study and of its sweeping conclusions about the general physics of cuprates unless the authors accompany their data with detailed modeling proving a direct link between the paramagnon data and $T_{c,max}$ – not only for Hg1201 and Hg1212 but for all other families. I believe that a simple correlative study of this kind, and one lacking perfect correlation as evidence by Figure 4, without mechanistic understanding does not hold the burden of proof for an important question such as the one surrounding the pairing mechanism for Cooper pairing in the CuO₂ planes.

I hope the authors can reconsider the formulation of their work and focus on the experimental data at hand, and independently from wider claims that cannot be fully substantiated.

I include a few remarks (some of which are formulated as questions) which I hope the authors will find useful in revising their manuscript:

- The authors are focusing on the correlation between the superconducting critical temperature and the energy of spin fluctuations. However, if the latter are hypothesized to be responsible for pairing, then what about the correlation with the temperature at which pairs begin to form, which some models and prior studies purport to be the pseudogap temperature? Is there any correlation between J and T^* ? It would be interesting to see a plot of the likes of Fig. 4 for these two quantities.

- If T_c is correlated to the paramagnon energy and the former is strongly doping dependent, shouldn't the latter be as well? I was hoping the authors to elaborate on this aspect, also in the manuscript.

- Reducing the comparison between the broad spectral features of the paramagnons in Hg1201 and Hg1212 to the difference in the centroid of a DHO fit seems quite an oversimplification, especially in a regime where the damping coefficient is of the same order of magnitude as the resonance energy. For instance, one might wonder what would be the additional role of the spectral weight distribution and absolute value of the spin susceptibility in a model calculation of T_c . How can one be so sure that the only relevant parameter is the DHO central energy? (this comment is related to my earlier call for a quantitative model of the connection between J and

T_{c,max} if the authors want to go down this route and make broad statements)

- On what grounds do the authors adopt linear spin-wave theory to model the dispersion of paramagnon frequency vs. momentum? Does linear spin-wave theory hold when magnons cease to be well-defined single-particle excitations of the system? (it seems to be the case here as signaled by the large damping factor)

- It seems fair that electron-doped cuprates (whose spin excitations have been similarly measured using RIXS and INS) shall be added to Figure 4, since superconductivity and magnetism arise from the very same ingredients and interactions as the hole-doped cuprates.

- The role of period-4 charge correlations as a hindrance to superconductivity does not sound convincing. For instance, in Bi2201 and CCOC, charge order is very short-ranged. Moreover, near-commensurate period-4 charge order is present also in Bi2212 and Hg1201, which are not in this subset of families. So this statement seems unsubstantiated, but I look forward to the authors elaborating on this.

- I was not aware that spin excitations exhibited an anomaly near the charge order wavevector in Bi2201 and La214. Is this a behavior that has been well characterized in all cuprate families, to the point that one can single out Bi2201 and La214 as the only families manifesting such anomaly? In any case, if the corresponding data could be reproduced in Supplementary Figure 10 together with the other RIXS panels, that would help the reader appreciate the existence of such anomaly.

Reviewer #2:

Remarks to the Author:

Dear Dr. Wang

I carefully read the manuscript "Paramagnons and high-temperature superconductivity in a model family of cuprates" by Professor Li and colleagues. In this paper the authors examine, in the cleanest possible family of cuprates, the relations between the critical temperature T_c and the super-exchange parameter J as measured by resonance inelastic x-ray scattering (RIXS) and Raman scattering. They find positive correlation between the two parameters in both techniques. They then put their values of J in a broad context of other cuprates. They find a global positive trend, and explain why some cuprate families fall out of this trend. I think that this is an excellent and important paper in the field of Cuprates superconductivity and it should be published without delay. However, I do have some comments for the authors to consider.

1) The authors motivate their research by suggesting that there are contradicting results in the literature and give reference 11, 12 and 14. The contradiction of Ref. 11 (T_c decreases with increasing J) was resolved by the same group leader in PRB 100, 144512 (2019). Ref. 12 agrees with the notion that T_c grows with increasing J. The contraction of Ref. 14 is irrelevant since the authors compare different cuprate families. If one compares arbitrary cuprates there are contradicting results even after the present work as presented in Fig. 4. Comparing different families is not a fair game.

2) The authors write "Even though a positive correlation between T_c and parent compounds' antiferromagnetic ordering temperature (T_N) has been found in some cases [10]..." Actually, the authors of reference 10 found correlation between T_c and J.

3) The authors write "Spectroscopic determination of the associated energies suggests a much weaker correspondence [12]". In the present manuscript there is also disagreement between the correspondence of two different spectroscopic techniques: RIXS and Raman. Although the authors justify this disagreement, the experimental reality is that it is there.

Therefore, I believe that trying to create a mirage of contradicting result in the past as a motivation for the work is not working. The "positive correlation" is universal within one cuprate family. The motivation to read this paper for me is: excellent and well characterized crystals with a

huge variation in T_c , with minute crystallographic variations. All three properties mount to the best experiment ever.

Another disturbing aspect of the manuscript is that the data of Ref. 10 is not included in Fig. 4 and no explanation is provided for this omission. It gives a feeling that the authors are hiding something, especially that CDW have been characterized for the samples of Ref. 10 (Bluschke PRB 100 035129 (2019)) and they could be discussed on the same footing as all samples in the Fig. 4.

To summarize, this is a well written and important paper adequate for nature communications, which should be published with minor but important corrections.

Reviewer #3:

Remarks to the Author:

This manuscript by L. Wang et al. reports high-resolution RIXS measurements of the paramagnon spectrum in the Hg-based cuprates $\text{HgBa}_2\text{CuO}_{4+d}$ and $\text{HgBa}_2\text{CaCu}_2\text{O}_{6+d}$. The scattering results are complemented by electronic Raman scattering spectra at several temperatures. The integration of these data allows the authors to speculate about a correlation between the superexchange interaction and the maximum superconducting T_c across several cuprate families.

The data comes from beamline I-21 of Diamond Light Source, which is one of the best RIXS instruments worldwide, and the experimental team has a strong track record in the study of magnetic fluctuations in the copper oxides. The work clearly underwent some revision, and the data analysis is solid. However, the manuscript does not seem to provide a novel message, besides measuring the paramagnons of yet another cuprate compound, and got me confused in several points.

I will start by commenting on the novelty. The paper makes a connection between the superexchange energy of the magnetic fluctuations, the superconducting T_c and the pair breaking peak in the electronic Raman spectrum. Based on empirical observations, the authors' main message is that a higher T_c correlates with a higher superexchange coupling J . In their words "Our result strongly suggests that the Cooper pairing is mediated by the paramagnons". In my opinion, this is already known, to some extent, thanks to seminal work by some of the authors, such as Le Tacon et al. Nature Phys 7, 725 (2011). Over the last ten years, paramagnons have been measured in many other cuprate families and this work provides other two examples. Hence, is this work superior or more impactful than other comprehensive studies such as PRB 97, 155144 and PRB 98, 144507? I am not sure.

Also, I have questions about the meaning of figure 4d. First, it is unclear that the T_c and J should be linearly correlated. I can see that in a BCS picture T_c is proportional to the boson energy, but Fig. 4c of Nature Phys 7, 725 shows a nonlinear dependence between the two quantities. How are these pictures mutually consistent? Also, one could argue whether a linear correlation would be visible after removing the guide to eye, but I will leave this discussion to the editor and other reviewers. Secondly, how accurate is the collapse of all these cuprate families on a single J axis? Some of the authors have previously emphasized that the spin fluctuation spectrum is best captured by including more than one exchange term, the cyclic ring exchange being often a very large correction (Peng et al. Nature Phys 13, 1201 (2017)). While being a very appealing picture, I am not sure that things are so simple as the authors articulate.

In addition to these issues, I am also puzzled about why the authors do not see a bilayer magnon splitting in the Hg1212 data. One would expect to see an optical and acoustic spin wave dispersion. Further, spin excitation splitting in bilayer and trilayer compounds should be accounted for when compiling Fig. 4d.

Also, did the authors perform self-absorption corrections on the RIXS spectra?

Lastly, the paper contains some statements such as "Cuprate superconductors have critical temperatures (T_c) on the order of a hundred kelvin, which is high by common standard but still "low" compared to the electronic energy scales". All superconductors have a T_c which is low

compared to hopping, Coulomb repulsion or Fermi energy E_F . This is even more the case of conventional superconductors at weak coupling, due to the exponential dependence on the coupling strength in the T_c formula. In some sense, the cuprates are high-temperature superconductors because their T_c/E_F ratio is rather large with respect to the one of conventional metals. Why would this be a distinctive cuprate feature?

Response to Reviewer #1's comments

"I have read the manuscript "Paramagnons and high-temperature superconductivity in a model family of cuprates" by Wang et al. with interest. The authors report an X-ray and Raman scattering study of the energy scales of the spin excitations and superconducting gap in the family of Hg cuprates. The authors purport that the observed correlation between the paramagnon energy, and correspondingly the Cu-Cu nearest-neighbor exchange interaction J , and the superconducting critical temperature is indicative, or perhaps strongly suggestive, of spin-mediated Cooper pairing in the cuprates.

Needless to say, what is the nature of the excitations and interactions mediating the formation of Cooper pairs in copper oxide high- T_c 's is a long-standing question that is central to the understanding of unconventional superconductivity. The authors sought to provide fresh insights to this question. They adopt an approach which has been followed in prior inelastic neutron scattering studies of spin excitations in the cuprates – that is, modeling the spin dynamics to extract an effective magnetic interaction scale (J) to compare to energy (Δ_{SC}) or temperature (T_c) scales relevant for superconductivity. The main novelty is that they focus on two cuprate families (Hg1201 and Hg1212) that had not been investigated very systematically and finish to compile the new data together with historical data to extract trends and correlations, especially between J and $T_{c,max}$. I enjoyed reading the study and learning about the new analysis of RIXS data in the Hg-cuprate family."

Our response: We thank the Reviewer for the time spent reviewing our manuscript and for the nice summary of our work. The Reviewer appreciates very well the scientific question that we set out to address, as well as the position of our work among related results in the literature.

"I personally think that the study would be worth of publication in Nature Communications on the sole standing of the present data, without invoking broad conclusions which, to my reading, sound somewhat speculative. I would be happy to endorse (after revisions – see remarks below) this manuscript for publication if the authors take this route. However, I cannot support publication of this study and of its sweeping conclusions about the general physics of cuprates unless the authors accompany their data with detailed modeling proving a direct link between the paramagnon data and $T_{c,max}$ – not only for Hg1201 and Hg1212 but for all other families. I believe that a simple correlative study of this kind, and one lacking perfect correlation as evidence by Figure 4, without mechanistic understanding does not hold the burden of proof for an important question such as the one surrounding the pairing mechanism for Cooper pairing in the CuO₂ planes.

I hope the authors can reconsider the formulation of their work and focus on the experimental data at hand, and independently from wider claims that cannot be fully substantiated."

Our response: We appreciate the Reviewer's candid and critical expression of the limitation of our work in its interpretations (namely, we suggested spin-mediated pairing and made

speculations on a few fronts) while acknowledging that the strength of our experimental data alone warrants publication in *Nature Communications*.

We have decided to follow the Reviewer's advice. In our revised manuscript (starting from the abstract and throughout the main text), we have focused on presentation of our experimental data and their immediate implications, and left out or deemphasized remarks on related but more distant subjects. In particular, when discussing the summary data in our Fig. 4d, we now explicitly point out how our new data help reveal the "correlative" relation between T_c and J , as acknowledged by the Reviewer, and how the results are still "lacking perfect correlation", as correctly noted by the Reviewer. We do keep a speculative note towards the end, but we have made it clear where the speculation starts and made it brief, mainly to point out a few connections to recent developments.

We believe that our revised manuscript will be seen as appropriate by the Reviewer for publication in *Nature Communications*. In the following, we respond to the Reviewer's specific comments and suggestions.

"-The authors are focusing on the correlation between the superconducting critical temperature and the energy of spin fluctuations. However, if the latter are hypothesized to be responsible for pairing, then what about the correlation with the temperature at which pairs begin to form, which some models and prior studies purport to be the pseudogap temperature? Is there any correlation between J and T^* ? It would be interesting to see a plot of the likes of Fig. 4 for these two quantities."

Our response: We thank the Reviewer for raising this important point and suggesting we make a comparison to T^* . We are aware of discussions of pre-formed Cooper pairs and their relationship to the pseudogap behaviour. To minimize ambiguity, we have used T_{pair} to indicate the associated onset temperature in our revised manuscript. There are actually two seemingly different types of experimentally observed T_{pair} . Because their values are rather different, we have refrained from plotting the reported data, as that would require us to discuss how we choose the data for plotting. Instead, we have added a new paragraph:

[main text, starting on page 7]

The cuprates are well-known to have the Cooper pairs formed without phase coherence above T_c due to the low superfluid density^{6,7}. The pair formation temperature (T_{pair}) above T_c is hence an alternative measure of the pairing strength. However, values of T_{pair} vary widely in the literature depending on the method of observation: By tracing the temperature evolution of gap formation (presumably due to pairing), angle-resolved photoemission spectroscopy revealed T_{pair} of Bi2201 and Bi2212 to be between 120 to 150 K in spite of their very different T_c ³⁷. Other techniques also supported similar T_{pair} in the Bi-family, YBCO and La214³⁸⁻⁴⁰. According to these results, the above compounds have similar T_{pair} (even though some of them have distinct T_c), and we note that they also possess similar J in Fig. 4d. In contrast, torque magnetometry suggested a universal and rather narrow temperature range of superconducting fluctuations (only a few Kelvins) above T_c ^{41,42}. In this definition, T_{pair} would closely track T_c , so

the maximal T_{pair} of a given family would have a similar relation to J as $T_{c,\text{max}}$. We therefore conclude that, despite on-going debates about T_{pair} in the literature (see Supplementary Table 5), the relationship between T_{pair} and J can be expected to remain consistent with the empirical trend revealed in Fig. 4d.

“If T_c is correlated to the paramagnon energy and the former is strongly doping dependent, shouldn't the latter be as well? I was hoping the authors to elaborate on this aspect, also in the manuscript.”

Our response: We thank the Reviewer for asking us to clarify this point. No, the paramagnon energy appears to be an approximately fixed value in a given cuprate, in spite of the fact that T_c can of course change as a function of doping. In particular, the parent compounds are antiferromagnets with large magnon energy but zero T_c . While this is an empirical fact, the physical reason is that the paramagnon energy ($\sim 2J$ at the zone boundary) in a given cuprate is determined by the electronic structure of the doped Mott (charge-transfer) insulator, t and U in particular, which harbour in the parent compound. These electronic-structure parameters are understood to be doping-independent. Following the Reviewer's suggestion, we have made the following revision, which also explains why our comparative study is between Hg1201 and Hg1212, rather than for different doping of one of them:

[main text, starting on page 2]

The pursuit for an experimental energy correspondence between pairing and the paramagnons has proved challenging. In the generic p - T phase diagram¹⁻³ of the cuprates, where T is temperature and p is the doping level, the decrease of T_c away from optimal doping is believed to be due to reduced superfluid density^{6,7}, whereas the zone-boundary paramagnon energies are nearly independent of p within a given cuprate^{5,8,9,10}. Usage of a tuning knob other than doping, such as chemical¹¹ and applied pressures¹², is thus required to investigate the correlation between the paramagnon energy and T_c

“Reducing the comparison between the broad spectral features of the paramagnons in Hg1201 and Hg1212 to the difference in the centroid of a DHO fit seems quite an oversimplification, especially in a regime where the damping coefficient is of the same order of magnitude as the resonance energy. For instance, one might wonder what would be the additional role of the spectral weight distribution and absolute value of the spin susceptibility in a model calculation of T_c . How can one be so sure that the only relevant parameter is the DHO central energy? (this comment is related to my earlier call for a quantitative model of the connection between J and $T_{c,\text{max}}$ if the authors want to go down this route and make broad statements)”

Our response: We agree with the Reviewer that our single-parameter definition (namely, by the DHO central energy, and then further by J , see the next response) of the paramagnon energy is an approximation. However, the paramagnon energy difference between Hg1201 and Hg1212 is an evident experimental fact according to the raw experimental data shown in our Figs. 1-2. One does not need to perform any data fitting to see that.

As it is not our intention to formulate a quantitative model to explain or propose a physical connection between J and $T_{c,max}$, we have used DHO fits simply to extract a representative value (a single approximate number) of the antiferromagnetic coupling strength for every cuprate in our Fig. 4. To this end, we believe that the DHO model is a commonly accepted one in the literature. In particular, this model is known to yield only weakly doping-dependent characteristic energy (the propagation energy) near the magnetic Brillouin zone boundary, and is hence ideal for our purpose here. To further justify our chosen approach, we have modified a sentence, which now reads:

[main text, starting on page 4]

Since visual comparison may be biased by the effect of damping, here we consider the ratio (~130%) associated with the propagation energy near the zone boundary (0.5, 0) physically more meaningful, because the propagation energy there is experimentally and theoretically known to change little with doping^{5,8-10,26}, so it plausibly reflects the intrinsic magnetic energy scale for a given cuprate.

We agree with the Reviewer that discussion of spectral weight distribution and absolute values of the spin susceptibility will be important for quantitative modelling, which falls beyond the scope of our present work.

“On what grounds do the authors adopt linear spin-wave theory to model the dispersion of paramagnon frequency vs. momentum? Does linear spin-wave theory hold when magnons cease to be well-defined single-particle excitations of the system? (it seems to be the case here as signaled by the large damping factor)”

Our response: The Reviewer makes a correct point that, strictly speaking, the spin-wave approach implies the existence of long-range magnetic order, which does not exist in superconducting cuprates including our samples. However, the cuprates are layered antiferromagnets, and even when long-range magnetic order is destroyed by thermal disorder or doping, sufficiently well-defined antiferromagnetic correlations still exist in the CuO₂ layers, and they support the existence of paramagnon excitations whose frequency vs. momentum dispersion in two dimensions closely resembles that in the magnetically ordered parent compounds described by linear spin-wave theory.

To clarify the fact that the spin-wave theory analysis is merely for extracting the paramagnon energy near the zone boundary (which amounts to $2J$) for comparison among different cuprates, we have modified the following sentences in our revision:

[main text, starting on page 2]

The paramagnon energy is commonly modelled and discussed in terms of the antiferromagnetic coupling strength (J) in the CuO₂ layers^{5,8,10}, which affects the magnetic ordering temperature (T_N) in parent compounds.

[main text, starting on page 7]

It is therefore reasonable to believe that J can vary considerably according to the crystal and electronic structures. Since J is the leading magnetic interaction term, this provides a rationale for our approximate description of the paramagnon spectra using a single parameter J .

[main text, starting on page 7]

Using J as a simplified and unified description of the paramagnon energy measured with inelastic neutron scattering and RIXS (detailed in Supplementary Fig. 10), we summarise the result and the different families' maximal value of T_c ($T_{c,max}$, at optimal or the best available doping) in Fig. 4d. The quoted values of J , obtained by linear spin-wave theory fitting (see Methods), are mainly constrained by the observed energies of zone-boundary (para)magnons, and a similar trend between the latter and $T_{c,max}$ is shown in Supplementary Fig. 11. This direct spectroscopic origin of the extracted J values renders it a simplified yet reliable quantity for comparison among the different families, some of which (including Hg1201 and Hg1212) lack parent compounds.

[Methods, starting on page 23]

For doped systems, since spin excitations hold collective nature up to the optimal doping²⁵ and its high-energy dispersion in the underdoped sample is similar to that of parent compound even with the damping^{8,47}, the LSWT is considered still valid to fit the high-energy paramagnon in underdoped samples.

“It seems fair that electron-doped cuprates (whose spin excitations have been similarly measured using RIXS and INS) shall be added to Figure 4, since superconductivity and magnetism arise from the very same ingredients and interactions as the hole-doped cuprates.”

Our response: We thank the Reviewer for the suggestion, and have added an electron-doped cuprate NCCO to Fig. 4, Supplementary Figs. 10 and 11, and Supplementary Tables 3-4. The associated discussions in the main text have been accordingly updated.

“The role of period-4 charge correlations as a hindrance to superconductivity does not sound convincing. For instance, in Bi2201 and CCOC, charge order is very short-ranged. Moreover, near-commensurate period-4 charge order is present also in Bi2212 and Hg1201, which are not in this subset of families. So this statement seems unsubstantiated, but I look forward to the authors elaborating on this.”

Our response: We appreciate the Reviewer's advice at the beginning of the report that we should deemphasize speculations to let the strength of our experimental data stand out. In our revision, we have simplified the ending (speculative) discussions to a single paragraph, where we no longer mention period-4 charge correlations:

[main text, starting on page 9]

We end our discussion with a speculative note motivated by the outliers in Fig. 4d. Their relatively low $T_{c,max}$ could be attributed to detrimental effects on T_c including disorder¹⁷, competing order¹ (e.g., stripe order in La214) and small hopping range of conduction electrons^{43,44}. Meanwhile, the reason behind the outliers might be related to a recent proposal that there is an optimal coupling strength between charge carriers and pair-mediating bosons in weak-coupling superconductors^{45,46}, where too strong couplings promote charge-order formation rather than superconductivity. We notice that La214, Bi2201 and NCCO have been reported to exhibit pronounced anomalies in their paramagnon spectra near a wave vector where charge correlations have been observed^{16,47,48}. Such anomalies hint at strong couplings between charge carriers and magnetic excitations in those systems^{49,50}, yet similar anomalies are nearly invisible in the Hg1212 and Hg1201 (Fig. 2d and Ref. 51). Whether this hints at a moderate coupling strength in Hg1201 and Hg1212 which favours high T_c warrants further experimental and theoretical studies.

“I was not aware that spin excitations exhibited an anomaly near the charge order wavevector in Bi2201 and La214. Is this a behavior that has been well characterized in all cuprate families, to the point that one can single out Bi2201 and La214 as the only families manifesting such anomaly? In any case, if the corresponding data could be reproduced in Supplementary Figure 10 together with the other RIXS panels, that would help the reader appreciate the existence of such anomaly.”

Our response: To our best knowledge, there have been reports of paramagnon anomalies near Q_{CDW} only in La214 (specifically LBCO-1/8), NCCO and Bi2201 (note that we have also added NCCO to our discussion regarding charge-paramagnon coupling in the revised manuscript). The paramagnon of LBCO-1/8 exhibits a profound softening around Q_{CDW} below T_{CDW} (Ref. [47]), whereas NCCO and Bi2201 exhibit intensity enhancement of paramagnons at Q_{CDW} , denoted as “intensity anomaly” (Refs. [16, 48]). For other families of cuprates including YBCO, Hg1201 and Hg1212, one might argue that there are some hints in the raw data, but the effects are generally much weaker.

Following the Reviewer’s suggestion, we have reproduced the paramagnon dispersion of LBCO-1/8 from Ref. 47 in Supplementary Fig. 10, in comparison to the dispersion in the parent compound. As for the “intensity anomaly” in NCCO and Bi2201, they were not presented in that figure since the effects require detailed temperature evolution to be revealed, and because Supplementary Fig. 10 aims at presenting and fitting paramagnon dispersions.

We thank the Reviewer again for their willingness to recommend our work for publication in *Nature Communications* and for all the constructive suggestions made.

Response to Reviewer #2's comments

"Dear Dr. Wang

I carefully read the manuscript "Paramagnons and high-temperature superconductivity in a model family of cuprates" by Professor Li and colleagues. In this paper the authors examine, in the cleanest possible family of cuprates, the relations between the critical temperature T_c and the super-exchange parameter J as measured by resonance inelastic x-ray scattering (RIXS) and Raman scattering. They find positive correlation between the two parameters in both techniques. They then put their values of J in a broad context of other cuprates. They find a global positive trend, and explain why some cuprate families fall out of this trend. I think that this is an excellent and important paper in the field of Cuprates superconductivity and it should be published without delay. However, I do have some comments for the authors to consider."

Our response: We thank the Reviewer for the time and effort spent reviewing our manuscript. We are very much pleased by the Reviewer's appreciation of the quality and importance of our work. We also thank the Reviewer for making constructive comments for us to further improve our manuscript.

"1) The authors motivate their research by suggesting that there are contradicting results in the literature and give reference 11, 12 and 14. The contradiction of Ref. 11 (T_c decreases with increasing J) was resolved by the same group leader in PRB 100, 144512 (2019). Ref. 12 agrees with the notion that T_c grows with increasing J . The contraction of Ref. 14 is irrelevant since the authors compare different cuprate families. If one compares arbitrary cuprates there are contradicting results even after the present work as presented in Fig. 4. Comparing different families is not a fair game."

Our response: We thank the Reviewer for drawing our attention to the new reference [PRB 100, 144512 (2019)] which resolved the earlier seemingly contradicting result of Ref. [11] (now [12]). Our intention was to give well-deserved credit to all these pioneering studies, while also indicating the outstanding difficulty that, because the 'dynamic range' of T_c and J variations were small, the earlier studies' quantitative clarity was somewhat limited. With the additional reference, we feel that it would be fair to rephrase our narrative of the previous studies into the following:

[main text, starting on page 2]

... The paramagnon energy is commonly modelled and discussed in terms of the antiferromagnetic coupling strength (J) in the CuO_2 layers^{5,8,10}, which affects the magnetic ordering temperature (T_N) in parent compounds. A positive correlation between T_c and J inferred from T_N has indeed been found previously¹¹. Yet, spectroscopic determination of the associated energies suggested a weaker correspondence¹³, and historically different impressions were once obtained^{12,14-16}. This is because major modifications of J are difficult to achieve, and they go along with structural modifications whose consequences are difficult to assess. ...

All references numbers are updated accordingly. The new reference [PRB 100, 144512 (2019)] is added as Ref. [14] above. The same revision also addresses the Reviewer's following comment:

"2) The authors write "Even though a positive correlation between T_c and parent compounds' antiferromagnetic ordering temperature (T_N) has been found in some cases [10]..." Actually, the authors of reference 10 found correlation between T_c and J ."

Our response: We agree with the Referee that the original conclusions of Ref. [10] (now [11]) were made about a correlation between T_c and J , the latter of which the authors inferred from a combination of T_N and temperature evolution of muon-spin-rotation results below T_N , which reveals the effective dimensionality of the antiferromagnetic coupling. Our revised summary of Ref. [11] above should more accurately reflect the nature of the previous finding.

"3) The authors write "Spectroscopic determination of the associated energies suggests a much weaker correspondence [12]". In the present manuscript there is also disagreement between the correspondence of two different spectroscopic techniques: RIXS and Raman. Although the authors justify this disagreement, the experimental reality is that it is there.

Therefore, I believe that trying to create a mirage of contradicting result in the past as a motivation for the work is not working. The "positive correlation" is universal within one cuprate family. The motivation to read this paper for me is: excellent and well characterized crystals with a huge variation in T_c , with minute crystallographic variations. All three properties mount to the best experiment ever."

Our response: The Reviewer's comment is fair. Indeed, different experimental methods can produce somewhat different quantitative results, and our RIXS + Raman combination is no exception of that. In our revised manuscript, we acknowledge the quantitative difference we have between our RIXS and Raman data:

[main text, starting on page 6]

... between the two systems, despite the fact that the relative difference (16%) in the peak energy determined from the spectral variation with temperature is smaller than the RIXS result, it matches precisely that of $2\Delta_{sc}$

We also realized that the word "much" (now left out) in the original sentence quoted by the Reviewer might have been too eye-catching. It was not our intention to create the impression that some of the previous work 'undid' the finding of even earlier work – our intention was to simply introduce the experimental challenge associated with the small variations in T_c and J . We appreciate the Reviewer's candid expression of concern over how our previous writing could have impressed a reader unfamiliar with the previous insightful studies, and hope that, with our revision, such unnecessary impression is adequately

mitigated. Indeed, as the Reviewer has correctly pointed out, making the desired variations (J and T_c) greater while containing unwanted variations to a minimal should be taken as the main advantage of our experimental work.

“Another disturbing aspect of the manuscript is that the data of Ref. 10 is not included in Fig. 4 and no explanation is provided for this omission. It gives a feeling that the authors are hiding something, especially that CDW have been characterized for the samples of Ref. 10 (Bluschke PRB 100 035129 (2019)) and they could be discussed on the same footing as all samples in the Fig. 4.

To summarize, this is a well written and important paper adequate for nature communications, which should be published with minor but important corrections.”

Our response: As the Reviewer had noticed, we did not include all previous estimations of J - T_c pairs into our Fig. 4. The reason was mostly from a technical point of view: we wanted to have only one data point for each of the major cuprate systems, and the data point's J value ought to come from a most direct spectroscopic method: inelastic neutron scattering or resonant inelastic X-ray scattering. This is not only to simplify the viewgraph, but also to set a uniform standard for the displayed values, such that a quantitative comparison becomes the most reliable.

We appreciate the fact that by adding the 4 data points enabled by the study in Ref. [10] (now [11]) would only make our Fig. 4d look more impressive, and it is indeed our plan to show those combined results in future oral presentations of this work. Yet for the present manuscript, keeping only the neutron and RIXS data points significantly simplifies the logic flow, as otherwise, we would have to spend even more lines to explain, *e.g.*, why values of J inferred from Raman (the two-magnon peak) are not included even for systems that have their parent compounds available. But we agree with the Reviewer that when discussing the empirical result in Fig. 4d, it is a good opportunity to make a connection to Ref. [10] (now [11]), hence we have made the following revision:

[main text, starting on page 7]

... Importantly, our new Hg1201 and Hg1212 result extends the data range both horizontally and vertically in Fig. 4d, revealing an approximate linear relationship between $T_{c,max}$ and J that includes a total of eight different cuprates with relatively high $T_{c,max}$. The empirical observation is consistent with a former study using chemical substitution to tune both J (estimated from T_N and inter-layer coupling) and T_c ¹¹, but a broader data range and many more compound families are covered here.

To end our response, we wish to thank the Reviewer again for the many delightful compliments of our work along with the constructive suggestions.

Response to Reviewer #3's comments

"This manuscript by L. Wang et al. reports high-resolution RIXS measurements of the paramagnon spectrum in the Hg-based cuprates $\text{HgBa}_2\text{CuO}_{4+d}$ and $\text{HgBa}_2\text{CaCu}_2\text{O}_{6+d}$. The scattering results are complemented by electronic Raman scattering spectra at several temperatures. The integration of these data allows the authors to speculate about a correlation between the superexchange interaction and the maximum superconducting T_c across several cuprate families.

The data comes from beamline I-21 of Diamond Light Source, which is one of the best RIXS instruments worldwide, and the experimental team has a strong track record in the study of magnetic fluctuations in the copper oxides. The work clearly underwent some revision, and the data analysis is solid. However, the manuscript does not seem to provide a novel message, besides measuring the paramagnons of yet another cuprate compound, and got me confused in several points."

Our response: We thank the Reviewer for the time and effort spent reviewing our manuscript. The Reviewer made an accurate summary of our experimental result, and we appreciate the Reviewer's recognition of the technical correctness of our work. The Reviewer's main reservation was about "a novel message", and we believe that it arose partly from our previous version's highlighted speculation about magnetically-mediated Cooper pairing. Indeed, we are not the first to speculate "a correlation between the superexchange interaction and the maximum superconducting T_c ."

What we joined previous researchers in speculating is one of the most obvious things to speculate about in the high- T_c cuprates – because superconductivity arises from parent compounds which are antiferromagnets, it is natural to expect antiferromagnetism to help with Cooper-pair formation. While the expectation is shared by many, it is fair to say that a consensus has not been reached. More importantly, it is a very different matter to experimentally validate the speculated scenario. Our work's main novelty and importance reside in this latter part, namely, the design of our experiment is unprecedented, and the experiment yielded unprecedentedly clear-cut data to advance the experimental front regarding a correlation between the strength of the antiferromagnetic interactions and the maximal T_c . This view is shared by our other reviewers. In this regard, we disagree with the Reviewer that we merely measured "yet another cuprate compound." We respond to the Reviewer's specific comments in the following.

"I will start by commenting on the novelty. The paper makes a connection between the superexchange energy of the magnetic fluctuations, the superconducting T_c and the pair breaking peak in the electronic Raman spectrum. Based on empirical observations, the authors' main message is that a higher T_c correlates with a higher superexchange coupling J . In their words "Our result strongly suggests that the Cooper pairing is mediated by the paramagnons". In my opinion, this is already known, to some extent, thanks to seminal work by some of the authors, such as Le Tacon et al. *Nature Phys* 7, 725 (2011). Over the last ten

years, paramagnons have been measured in many other cuprate families and this work provides other two examples. Hence, is this work superior or more impactful than other comprehensive studies such as PRB 97, 155144 and PRB 98, 144507 ? I am not sure.”

Our response: We thank the Reviewer for pointing out the need to clarify what aspects of our present work stands out among the existing literature. We would like to begin by quoting a remark from our Reviewer #2: *“The motivation to read this paper for me is: excellent and well characterized crystals with a huge variation in T_c , with minute crystallographic variations. All three properties mount to the best experiment ever.”*

A key experimental design in our work is to use a tuning knob other than doping to change J and $T_{c,max}$. This makes our work entirely different in design from the two previous doping-dependence studies [PRB 97, 155144] and [PRB 98, 1445507] mentioned by the Reviewer. Moreover, our study is carried out in a model family of cuprates, and it expands the available data range of J vs. $T_{c,max}$ (in Fig. 4d). Both the cleanness of the Hg-family and the data-range expansion are crucial for revealing the intrinsic positive correlation between J and $T_{c,max}$. In our revision, we emphasize these merits of our result, including using a devoted paragraph:

[main text, starting on page 7]

... Importantly, our new Hg1201 and Hg1212 result extends the data range both horizontally and vertically in Fig. 4d, revealing an approximate linear relationship between $T_{c,max}$ and J that includes a total of eight different cuprates with relatively high $T_{c,max}$. The empirical observation is consistent with a former study using chemical substitution to tune both J (estimated from T_N and inter-layer coupling) and T_c ¹¹, but a broader data range and many more compound families are covered here.

[main text, starting on page 8]

Admittedly, there are distinct outliers from the linear trend in Fig. 4d, hence one should use cautions and not to prematurely interpret the trend as suggesting a causality relation between J and $T_{c,max}$. In fact, without our Hg-family data points, the approximate linear trend would have become much less evident even among the high- $T_{c,max}$ cuprates. Specifically, considering the data points of single- and bi-layer cuprates including $Tl_2Ba_2CuO_{6+\delta}$ (Tl2201), YBCO, $YBa_2Cu_4O_8$ (Y124), $NdBa_2Cu_3O_{6+\delta}$ (NBCO) and Bi2212, one might obtain the impression that J can vary by over 50% without affecting $T_{c,max}$ by more than 10%. Some of these compounds' further comparison to the low- $T_{c,max}$ ones including Bi2201, $La_{2-x}(Sr,Ba)_xCuO_4$ (La214), $Ca_{2-x}Na_xCuO_2Cl_2$ (CCOC) and $Nd_{2-x}Ce_xCuO_4$ (NCCO) could even suggest an anti-correlation between J and $T_{c,max}$. Meanwhile, a comparison within the bismuth-family of cuprates, Bi2201, Bi2212 and $Bi_2Sr_2Ca_2Cu_3O_{10+\delta}$ (Bi2223), could suggest that $T_{c,max}$ increases by a factor of three without much variation in J , which would appear to indicate that the number of multiple CuO_2 layers in the primitive cell is much more relevant to the variation in $T_{c,max}$ than the strength of magnetic interactions. Our Hg1201 and Hg1212 data points are thus highly elucidating in such an elusive situation, as the Hg-family is believed to suffer the least from

material-specific detrimental factors on T_c ¹⁷⁻¹⁹, which makes them more likely to reveal intrinsic correlations.

The importance of our results has been recognized in two recent theoretical studies: [Kowalski et al., PNAS 118, 40 (2021)] and [Cui et al., arXiv:2112.09735]. Both studies cited our result prominently, in abstract and multiple figures. This clearly shows the impact of our study. In our revised concluding remark, we draw our readers' attention to these two recent developments:

[main text, starting on page 9]

... Our finding brings fresh insight into the discussion of the role of antiferromagnetic coupling strength for affecting T_c in the cuprates^{36,52}, and renders the Hg-family of cuprate ...

“Also, I have questions about the meaning of figure 4d. First, it is unclear that the T_c and J should be linearly correlated. I can see that in a BCS picture T_c is proportional to the boson energy, but Fig. 4c of Nature Phys 7, 725 shows a nonlinear dependence between the two quantities. How are these pictures mutually consistent?”

Our response: We believe that the Reviewer's impression about Fig. 4c of [Nat. Phys. 7, 725] was inaccurate. The aim of that figure was to answer the following question: If only a limited part of the full magnetic excitation spectrum would contribute to pairing, how high T_c can reach as a function of the part's cutoff momentum (bottom x -axis) and energy (top x -axis, note the non-linear scale, we believe that there is little “nonlinear dependence” here)? This is a very different context from ours. In [Nat. Phys. 7, 725], it was pointed out that inelastic neutron scattering was only able to observe part of the full excitation spectrum, and thus, T_c would be underestimated if only that part is assumed to contribute.

In our case, we show that the full experimental RIXS spectra of Hg1201 and Hg1212 are basically identical apart from a global difference in the energy scale, as emphasized in our abstract. In such a case, T_c would be proportional to the energy scale of the overall boson energy. There is no contradiction between this understanding and the previous Fig. 4c of [Nat. Phys. 7, 725].

“Also, one could argue whether a linear correlation would be visible after removing the guide to eye, but I will leave this discussion to the editor and other reviewers.”

Our response: We thank the Reviewer for the fair remark. In fact, we think it is important to appreciate the fact that the empirical correlation in our Fig. 4d is far from being perfect – by virtue of existence of extrinsic factors affecting T_c in the different cuprates.

In our revised manuscript, we devote an entire paragraph (quoted on the previous page of this response) to discuss the imperfection of the correlation in Fig. 4d, and point out a few alternative impressions if our Hg1201 and Hg1212 data points were absent from the plot. We believe that such imperfection exactly shows how our new data will be appreciated.

“Secondly, how accurate is the collapse of all these cuprate families on a single J axis? Some of the authors have previously emphasized that the spin fluctuation spectrum is best captured by including more than one exchange term, the cyclic ring exchange being often a very large correction (Peng et al. Nature Phys 13, 1201 (2017)). While being a very appealing picture, I am not sure that things are so simple as the authors articulate.”

Our response: We agree with the Reviewer that the actual spin fluctuation spectrum might require additional terms beyond the nearest-neighbor J to be accurately described. However, for the purpose of our work, namely, to relate the superconducting and the magnetic energy scales in different cuprates, a simplified picture is desired because J is always the leading interaction that controls the overall energy scale, whereas the additional terms would generally raise the paramagnon energies over some momentum range but decrease it over some other range, *i.e.*, the overall energy scale remains unchanged. This understanding is consistent with the picture put forward in [Peng et al., Nature Phys 13, 1201 (2017)] mentioned by the Reviewer. In our revised manuscript, we spell out the fact that our single- J picture is an approximation, yet it is a reasonable approach to take for the goal of our study:

[main text, starting on page 7]

It is therefore reasonable to believe that J can vary considerably according to the crystal and electronic structures. Since J is the leading magnetic interaction term, this provides a rationale for our approximate description of the paramagnon spectra using a single parameter J .

[main text, starting on page 7]

Using J as a simplified and unified description of the paramagnon energy measured with inelastic neutron scattering and RIXS (detailed in Supplementary Fig. 10), we summarise the result and the different families' maximal value of T_c ($T_{c,max}$, at optimal or the best available doping) in Fig. 4d. The quoted values of J , obtained by linear spin-wave theory fitting (see Methods), are mainly constrained by the observed energies of zone-boundary (para)magnons, and a similar trend between the latter and $T_{c,max}$ is shown in Supplementary Fig. 11. This direct spectroscopic origin of the extracted J values renders it a simplified yet reliable quantity for comparison among the different families, some of which (including Hg1201 and Hg1212) lack parent compounds.

“In addition to these issues, I am also puzzled about why the authors do not see a bilayer magnon splitting in the Hg1212 data. One would expect to see an optical and acoustic spin wave dispersion. Further, spin excitation splitting in bilayer and trilayer compounds should be accounted for when compiling Fig. 4d.”

Our response: We do not see a bilayer magnon splitting in the Hg1212 data because the largest splitting is expected at 2D momentum transfer near zero, and it corresponds to a scattering geometry where specular reflection from the sample surface strongly contaminates the RIXS spectra. Incidentally, such splitting was also not resolved in YBCO in previous RIXS studies including [Nature Phys 7, 725 (2011)]. Away from the 2D zone center,

the two branches quickly merge together in energy, and by the time the dispersion reaches the zone boundary where the highest energy is observed, the branches become completely indistinguishable with our current energy resolution. For this reason, there is no reason to expect the splitting (near zone boundary) would affect the display in Fig. 4d in any practical sense – the data points would simply fall on top of each other. In our revision, we have added a brief mention of the fact that we do not resolve the bilayer splitting, and refer readers to technical details in Methods:

[main text, starting on page 5]

... Although magnon branch-splitting is not resolved here in Hg1212 (see Methods), magnetic coupling between the adjacent CuO₂ layers in a bi-layer cuprate is known to be only about 10 meV, which has little effect on the (para)magnon energy far away from the zone centre²⁷. Thus, the observed energy increase must originate in the strength of interactions within the CuO₂ layers. ...

[Methods, starting on page 18]

...The detector was placed at 154°, and the polarization state of the scattered phonons was not analysed. For this reason, overwhelming charge-scattering dominations (the elastic line) were observed in the spectra close to (0, 0) where too large uncertainties were obtained for the fitting results of paramagnon, so we only display scans at Q_{||} larger than (0.11, 0) and (0.07, 0.07). For the bi-layer compound, the optical and acoustic magnon overlap away from zone center⁵, therefore the magnon splitting of two branches in Hg1212 cannot be resolved in this setup. ...

“Also, did the authors perform self-absorption corrections on the RIXS spectra?”

Our response: We did not perform self-absorption corrections to our RIXS spectra. As explained in Methods, the only data treatment employed was the normalization of spectra taken at different momentum transfers according to the intensity of *d-d* excitations (this is a method commonly used in RIXS data analysis, see, *e.g.*, Refs. 5, 8-9 of our manuscript). We would like to stress that self-absorption correction would not change our conclusion at all, because it would bring similar correction to both Hg1201 and Hg1212.

“Lastly, the paper contains some statements such as “Cuprate superconductors have critical temperatures (T_c) on the order of a hundred kelvin, which is high by common standard, but still the T_c “low” compared to the electronic energy scales”. All superconductors have a T_c which is low compared to hopping, Coulomb repulsion or Fermi energy E_F. This is even more the case of conventional superconductors at weak coupling, due to the exponential dependence on the coupling strength in the T_c formula. In some sense, the cuprates are high-temperature superconductors because their T_c/E_F ratio is rather large with respect to the one of conventional metals. Why would this be a distinctive cuprate feature?”

Our response: We agree with the Reviewer that the previous opening sentence did not really make a distinctive case for the cuprates. We have revised it into the following:

[beginning of abstract]

Cuprate superconductors have the highest critical temperatures (T_c) at ambient pressure, yet a consensus on the superconducting mechanism remains to be established. ...

We would end our response by thanking the Reviewer again for the critical yet constructive comments, which we consider helpful to us preparing a revision of our manuscript. It is our opinion that the Reviewer had somewhat under-appreciated the importance and novelty of our work, and we hope that, by focusing more on the experimental results and making less speculations in our revision, we would manage to let the unprecedented “standing of the present data” (Reviewer 1) become clearer, and convince the Reviewer that our study warrants publication in *Nature Communications*.

Reviewers' Comments:

Reviewer #1:

Remarks to the Author:

The authors have thoroughly addressed all my remarks, both in the revised manuscript and in their response letter. I appreciate all their work and I am happy to fully endorse this manuscript for publication in Nature Communications.

Reviewer #2:

Remarks to the Author:

My concerns have been addressed in the revisions. In my opinion the manuscript is ready for publication.

Reviewer #3:

Remarks to the Author:

The authors provided a comprehensive response to the remarks/criticisms raised during the first round of review.

Furthermore, they have implemented a number of changes which improved the quality of the manuscript.

While they have addressed most of my doubts, I still have an issue about the collapse of all the data on a single J axis. In their response the authors state that "a simplified picture is desired because J is always the leading interaction that controls the overall energy scale,...". This is not always the case, and the cyclic exchange term can be substantial (see Peng et al. Nat Phys 2017 table 1). Perhaps this might provide an explanation for the outliers in the current plot, as the correlation between J and T_c might be a plane in the (J, J_c, T_{c_max}) space, rather than a line. I understand the appeal of such simplification, but I am afraid this is a case of over-simplification. Have the authors a quantitative reason to neglect J_c ? If so, they should articulate it.

Concerning the novelty, I still maintain that this is a work adding only two more points to a fairly established trend. However, I also see that two other reviewers expressed a different opinion and I will defer any further decision to the editor.

Response to Reviewers' comments

Reviewer #1: The authors have thoroughly addressed all my remarks, both in the revised manuscript and in their response letter. I appreciate all their work and I am happy to fully endorse this manuscript for publication in *Nature Communications*.

Reviewer #2: My concerns have been addressed in the revisions. In my opinion the manuscript is ready for publication.

Our response: We thank the Reviewers for the time spent on our revised manuscript. We are much delighted that they fully support our work for publication in *Nature Communications*.

Reviewer #3: The authors provided a comprehensive response to the remarks/criticisms raised during the first round of review. Furthermore, they have implemented a number of changes which improved the quality of the manuscript.

While they have addressed most of my doubts, I still have an issue about the collapse of all the data on a single J axis. In their response the authors state that "a simplified picture is desired because J is always the leading interaction that controls the overall energy scale,...". This is not always the case, and the cyclic exchange term can be substantial (see Peng et al. *Nat Phys* 2017 table 1). Perhaps this might provide an explanation for the outliers in the current plot, as the correlation between J and T_c might be a plane in the (J, J_c, T_{c_max}) space, rather than a line. I understand the appeal of such simplification, but I am afraid this is a case of over-simplification. Have the authors a quantitative reason to neglect J_c ? If so, they should articulate it.

Concerning the novelty, I still maintain that this is a work adding only two more points to a fairly established trend. However, I also see that two other reviewers expressed a different opinion and I will defer any further decision to the editor."

Our response: We are glad to learn that the Reviewer considers most of their previous concerns as having been sufficiently addressed.

We thank the Reviewer for understanding our reasoning of the simplification, and for their suggestion of articulating a quantitative reason to neglect cyclic exchange. In our J -only model, the neglect of the cyclic exchange term J_c can indeed lead to a slight underestimate of the actual nearest-neighbour superexchange term J , but this is minor because the maximal magnon energy $\omega(0.5, 0)$ is proportional to $J - J_c/10$ [Ref. 44]. Moreover, in the RIXS spectra, J_c is mainly reflected by the dispersion energy difference between momenta $(0.5, 0)$ and $(0.25, 0.25)$, and this difference is almost identical in our data between Hg1201 and Hg1212. (Although, we note that to accurately determine J_c , one should study antiferromagnetic parent compounds, which are unfortunately unavailable for Hg1201 and Hg1212.) The bottom line is that, while a variation of J_c may somewhat affect comparison among different families of cuprates under our simplified J -only model, it does not show

much difference between Hg1201 and Hg1212, so our main conclusion regarding the correlation between the single-parameter J and $T_{c,max}$ is unaffected by the simplification.

We have made the following revision to articulate the reason why J_c can be safely neglected in our current scope:

[Method, on page 15]

... In our model, the neglect of high-order interactions (for instance, cyclic exchange J_c) might lead to slight underestimation of the real nearest-neighbour exchange in CCO and La214⁴⁴, but a correction to those data points in Fig. 4 will not affect our conclusion. With the understanding that J_c is reflected by the energy difference between (0.5, 0) and (0.25, 0.25)⁴⁴, it is expected to be similar between our superconducting samples of Hg1201 and Hg1212 (Supplementary Table 1), yet we note that the most accurate value of J_c should be obtained from measurements of undoped samples which are not available at present. ...

Finally, we thank the Reviewer again for their time and effort assessing our manuscript, and especially for all the constructive suggestions.